# Ligands for Melanocortin Receptors: Beyond Melanocyte-Stimulating Hormones and Adrenocorticotropin

**DOI:** 10.3390/biom12101407

**Published:** 2022-10-01

**Authors:** Xiao-Chen Yuan, Ya-Xiong Tao

**Affiliations:** 1College of Animal Science and Technology, Anhui Agricultural University, Hefei 230061, China; 2Department of Anatomy, Physiology and Pharmacology, College of Veterinary Medicine, Auburn University, Auburn, AL 36849, USA

**Keywords:** neural melanocortin receptors, obesity, small molecule, pharmacoperone, drug repurposing

## Abstract

The discovery of melanocortins in 1916 has resulted in more than 100 years of research focused on these peptides. Extensive studies have elucidated well-established functions of melanocortins mediated by cell surface receptors, including MSHR (melanocyte-stimulating hormone receptor) and ACTHR (adrenocorticotropin receptor). Subsequently, three additional melanocortin receptors (MCRs) were identified. Among these five MCRs, MC3R and MC4R are expressed primarily in the central nervous system, and are therefore referred to as the neural MCRs. Since the central melanocortin system plays important roles in regulating energy homeostasis, targeting neural MCRs is emerging as a therapeutic approach for treating metabolic conditions such as obesity and cachexia. Early efforts modifying endogenous ligands resulted in the development of many potent and selective ligands. This review focuses on the ligands for neural MCRs, including classical ligands (MSH and agouti-related peptide), nonclassical ligands (lipocalin 2, β-defensin, small molecules, and pharmacoperones), and clinically approved ligands (ACTH, setmelanotide, bremelanotide, and several repurposed drugs).

## 1. Introduction

Melanocortin peptides, primarily consisting of α-, β-, and γ-melanocyte-stimulating hormones (α-, β-, γ-MSH) and adrenocorticotropin (ACTH), are elements of an ancient modulatory system [1]. Melanocortins are derived from tissue-specific posttranslational processing of the common precursor—proopiomelanocortin (POMC) [2]. The melanocortins are involved in pleiotropic physiological functions, including pigmentation, adrenal steroidogenesis, appetite regulation and energy homeostasis, energy metabolism, cardiovascular function, exocrine secretion, inflammation, immunomodulation, neuromuscular regeneration, sexual function, and temperature control [3,4,5,6,7,8]. Recent FDA approval of melanocortin-based drugs is expected to further promote the investigation of the melanocortin system and lead to new pharmacotherapies.

G protein-coupled receptors (GPCRs) have been tractable drug targets for decades, with over one-third of currently marketed drugs targeting GPCRs [9,10,11]. Of these, family A rhodopsin-like GPCRs are highly represented and continued drug discovery for this family of receptors may provide novel therapeutics for a wide range of diseases [12]. Melanocortin receptors (MCRs) are typical members of family A GPCRs. Extensive studies spanning several decades have elucidated the functions of melanocortins mediated by cell surface receptors named MSH receptor and ACTH receptor. In the early 1990s, with degenerate PCR cloning techniques, three additional related MCRs were cloned, named MC3R, MC4R, and MC5R, based on the chronological order of their characterization, with MSH receptor renamed as MC1R and ACTH receptor renamed as MC2R [1,3].

Among these receptors, MC3R and MC4R are expressed primarily in the central nervous system, and therefore referred to as the neural MCRs [13,14,15,16]. Analysis of mice lacking either *Mc3r* or *Mc4r* or both revealed that these two neural MCRs served important nonredundant roles in regulating energy homeostasis, with MC3R regulating feed efficiency and MC4R regulating both food intake and energy expenditure [17,18,19,20,21]. Targeting neural MCRs is emerging as a therapeutic approach for metabolic diseases and chronic inflammation, but until recently, drug development was limited by lack of selectivity and safety concerns [22,23,24].

Several endogenous ligands interact with the MCRs (Figure 1). Table 1 summarizes the pharmacological properties of these ligands at the human MCRs. Melanocortins have been studied for over a century, resulting in multiple drugs approved for treating several diseases [25,26,27]. This review focuses on the ligands for the two neural MCRs, including endogenous ligands (MSHs, agouti-related peptide (AgRP), and lipocalin 2), nonclassical ligands (β-defensin, small molecules, and pharmacoperones), and clinically approved ligands (ACTH, setmelanotide, and bremelanotide, and several repurposed drugs).

## 2. Endogenous Ligands

### 2.1. MSHs as Agonists

The endogenous agonists of neural MCRs are α-, β-, and γ-MSH, as well as ACTH, with the conserved tetrapeptide sequence His-Phe-Arg-Trp (HFRW) as the pharmacophore (Figure 1A), the minimally active sequence that possesses agonist activity in the classic frog and lizard skin bioassays [31,32]. Activation of the MCRs with MSH analogues indicated that the length of ACTH or MSH is different for a specific receptor subtype. Four amino acids (His-Phe-Arg-Trp) are the minimal MSH peptide for MC1R activation. The first 17 amino acids of ACTH are required for MC2R activation [33]. Four amino acids (His-Phe-Arg-Trp) in MSH are required for MC3R and MC5R activation [34]. Three amino acids (Phe-Arg-Trp) in MSH are required for MC4R activation [35]. Of the four major endogenous melanocortins, human MC1R and MC5R have the highest affinities for α-MSH, MC3R has the highest affinity for γ_1_-MSH, and MC4R has the highest affinity for β-MSH (Table 1) [28,29].

Pro-ACTH is cleaved by PC1 in the anterior pituitary corticotrophs to produce ACTH(1–39), which can be further processed through PC2 to produce ACTH(1–13)-NH_2_ and α-MSH [36]. ACTH is the only known endogenous ligand for MC2R, making it the sole agonist that stimulates all five MCRs. The full-length ACTH is 39 amino acids, of which ACTH(1–24) is hypothesized to be the functional domain. An anorectic effect was observed at 4 h after intracerebroventricular (ICV) administration of ACTH(1–24) at 0.05 μg/animal in mice and 10 μg/animal in rabbits [37]. A 4 μg/animal dose of ACTH(1–24) injected into the lateral ventricle or ventromedial hypothalamus also results in decreased food intake in rats [38].

The N-terminal 13 residues of ACTH are cleaved and modified through N-terminal acetylation and C-terminal carboxyamidation to produce α-MSH (Figure 1A). Acetylated α-MSH is more resistant to degradation than des-acetylated form [39]. At the human MCRs, α-MSH has been reported to possess single-digit nanomolar binding affinity only at the MC1R (Table 2). This peptide possesses nonselective subnanomolar to nanomolar potencies at the MC1R and MC3-5Rs (Table 3). Alanine scans of α-MSH have indicated the Met^4^, Phe^7^, Arg^8^, and Trp^9^ positions affect binding affinity and functional activity at the mouse MC1R and rat MC3R [40,41]. Expression of α-MSH in the central nervous system is predominantly in the hypothalamus, dispersed throughout the arcuate nucleus, the dorsomedial nucleus of the hypothalamus, and fibers in the medial preoptic, paraventricular, periventricular, and anterior hypothalamic nuclei [42,43]. ICV administration of α-MSH reduces food intake in rodents and intraperitoneal (IP) injection of α-MSH alters skin/hair coloration of humans and other mammals [44,45].

Unlike α-MSH, both termini of 22-residue β-MSH are unmodified (Figure 1A). Compared to the human MC3-5R, β-MSH has lower binding affinity at the MC1R (Table 2). Rodents lack the dibasic cleavage site for β-MSH and are therefore β-MSH-deficient [62]. However, human β-MSH reduces food intake in rats after ICV administration [63,64,65]. Although most studies on body weight regulation focused on α-MSH, β-MSH has also been shown to be important in the regulation of human energy homeostasis. The obesity phenotype in *POMC* mutation carriers resulting in β-MSH deficiency (Tyr5Cys, Tyr221Cys, and Arg236Gly) implies a significant effect of β-MSH on body weight [66,67,68], underscoring the importance of β-MSH as a physiologically relevant melanocortin ligand. In one study, human plasma β-MSH ranges from 20 to 110 ng/L [69]. In another study, cerebrospinal fluid samples obtained from 30 patients have a mean β-MSH concentration of 60.1 ng/L, compared with mean plasma concentration of 16.1 ng/L for normal adults [70]. Significant amounts of β-MSH were also observed in various regions of the brain [71].

The N-terminal domain of POMC contains three γ-MSH peptides: γ_3_-MSH (23-residue *N*-glycosylated peptide), γ_2_-MSH (N-terminal 12-residue cleaved from γ_3_-MSH), and γ_1_-MSH (N-terminal 11-residue cleaved from γ_3_-MSH with a C-terminal carboxyamidation). An alanine scanning of γ_2_-MSH indicated the importance of Met^3^, His^5^, Phe^6^, Arg^7^, and Trp^8^ for functional activity at the human MC3-5R, similar to residues in α-MSH that are important for activity [59]. γ-MSH is expressed in the pituitary and arcuate nucleus, as well as adrenal medulla [72,73,74,75]. Differences in receptor selectivity have been observed between species, where γ_2_-MSH is selective for the human MC3R over MC4R and MC5R, whereas there is no selectivity between the mouse MC3R and MC5R [59,76,77]. The >50-fold selectivity of three γ-MSHs for the MC3R over the MC4R led to several investigations of the role the MC3R might play in food intake regulation by administering γ-MSH ligands in vivo (Table 3). Interestingly, the γ-MSH sequence has been deleted from ancestral teleost *pomc* sequences [78], in parallel with the absence of the *mc3r* in some teleosts [79,80], providing support for coevolution of melanocortin ligand and the corresponding MCR.

### 2.2. AgRP as an Inverse Agonist

The MCRs are unique in having two endogenous antagonists, agouti (agouti-signaling protein, ASIP) and AgRP, the only endogenous GPCR antagonists known until 2018, when liver-expressed antimicrobial peptide 2 (LEAP-2) was identified as an endogenous antagonist of growth hormone secretagogue receptor 1a (GHS-R1a, ghrelin receptor) [81]. ASIP and AgRP contain a C-terminal tripeptide Arg-Phe-Phe (RFF) sequence postulated to be critical for binding and functional activity [82,83,84] (Figure 1B). Both proteins possess five C-terminal disulfide bonds, creating a similar cysteine knot conformation [82,84,85].

The active form of ASIP has been hypothesized to be ASIP(23–132), following cleavage of the N-terminal 22 residue signal peptide [85].This protein, produced by hair follicle, acts on melanocytes to inhibit α-MSH-induced eumelanin production by serving as an antagonist at the MC1R [86]. Indeed, ASIP is shown to be an inverse agonist for the MC1R [87]. Pharmacological studies demonstrated that ASIP is also a competitive antagonist at the MC4R, but did not interact with the MC3R [86]. A later study reported that ASIP inhibits the generation of cAMP stimulated by α-MSH at human MC1R and MC3-5R or by ACTH at human MC2R [53] (Table 3). Expression of ASIP is normally limited to the skin, where it affects pigmentation, while ubiquitous expression of ASIP causes other physiological functions, such as its expression in central nervous system resulting in obesity by inhibiting the function of MC4R [53,82,88].

AgRP consists of 112 amino acids (the gene product of 132 amino acids is processed by removal of the N-terminal 20-residue signal peptide). Despite these structural similarities, these two antagonists possess different pharmacological profiles at the MCRs. AgRP is not an antagonist at the MC1R but does interact with the centrally expressed MC3R and MC4R [51,82] (Table 3). Truncated and chimeric ASIP-AgRP ligands indicated that the C-terminal loop of ASIP is responsible for MC1R selectivity [89]. Ubiquitous expression of AgRP in the central nervous system results in increased food intake and obesity without altering pigmentation [82]. AgRP is predominantly expressed in the arcuate nucleus and binds to neural MCRs with nanomolar affinity [51,82]. In *Pomc*-null mice, administration of AgRP has delayed and long-lasting effects on energy balance and appetite control in a melanocortin signaling-independent way [90]. Recent studies in vivo using mouse model lacking neuronal MSH reported that basal intracellular cAMP levels in cells with MC3R or MC4R expression are decreased by AgRP administration [90], indicating AgRP functions as an inverse agonist at the constitutively active neural MCRs [91,92,93,94]. In addition, AgRP also acts as a biased agonist that activates the mitogen-activated protein kinase signaling pathway [94,95,96].

As in mammals, AgRP also plays critical roles in regulating food intake and energy homeostasis in teleosts [97], acting as an inverse agonist at neural MCRs of many fish [98,99,100,101,102,103]. We and others have shown that both Mc3r and Mc4r in fish have very high constitutive activities compared with human MC3R and MC4R, respectively [80,99,100,101,103,104,105,106,107,108,109,110,111]. Unlike mammals, two gene products (Agrp1 and Agrp2) of Agrp have been identified in several fish species, such as goldfish *Carassius auratus* [98], zebrafish *Danio rerio* [99], Atlantic salmon *Salmo salar* [112], seabass *Dicentrarchus labrax* [113], and pufferfish *Takifugu rubripes* [114] (reviewed in [115]). In seabass (Perciforme), long-term fasting increases hypothalamic expression of Agrp1 but decreases that of Agrp2 [113], suggesting an isoform-specific orexigenic action in fish.

### 2.3. Lipocalin 2

Lipocalin 2 was previously known as neutrophil gelatinase-associated protein exclusively secreted by adipose tissue (an adipokine) and associated with obesity and insulin resistance [116,117]. Recently, lipocalin 2 was shown to be also produced by osteoblasts, at levels much higher than white adipose tissue or other organs [54]. Recently, lipocalin 2 was identified as peripheral-derived novel potent polypeptide agonists for human MC1R, MC3R, and MC4R [54]. This peptide has been reported to possess nanomolar binding affinity at human MC1R, MC3R, and MC4R [54] (Table 2). Lipocalin 2 can activate MC4R in PVN neurons expressing neuropeptide Y Y1 receptor, suggesting that the osteoblast-derived peptide could cross the blood–brain barrier and suppresses appetite after binding to MC4R in hypothalamus [54,118]. In addition, loss- and gain-of-function experiments in mice demonstrate that lipocalin 2 maintains glucose homeostasis by enhancing brown fat activation and thermogenesis, inducing insulin secretion, and improving insulin sensitivity and glucose tolerance [54,117,119,120]. These studies demonstrated that lipocalin 2 links bone and the CNS for metabolic homeostasis of the whole body, and may be a key player for MC4R-mediated physiological functions in mammals.

## 3. Nonclassical Ligands

### 3.1. Defensin

β-Defensins are a rapidly evolving family of small secreted proteins thought to mediate responses to various and changing environmental stresses [121]. Mammalian β-defensins are expressed primarily in epithelial tissues, and usually induced in response to exposure to proinflammatory agents [122]. Initially recognized for their potential as “endogenous antibiotics,” some β-defensins can act as ligands for MCRs [123,124]. Canine β-defensin 3 (cBD103) solved a long-standing mystery regarding the dominant inheritance of black coats in domestic dogs through modulating melanocortin signaling [123]. A follow-up study conducted a structure–function analysis of the human orthologue of cBD103, human β-defensin 3 (hBD3), to determine the physicochemical properties of its high-affinity binding to MCRs [56] (Table 2). The unusual feature of melanocortin system is the modulation of the MCRs in the opposite direction by endogenous agonists (α-MSH, β-MSH, γ-MSH, and ACTH) and inverse agonists (ASIP and AgRP). Genetic and pharmacologic alstudies indicate that hBD3 acts as a neutral MCR antagonist capable of blocking the action of either stimulatory endogenous agonists such as α-MSH or inhibitory inverse agonists such as ASIP and AgRP [56]. Accordingly, central administration of hBD3 results in reduced food intake and decreased weight gain in rats [55,56]. Recently, human and mouse β-defensin 3 as well β-defensin 1 are shown to be full micromolar agonists at the MCRs, including MC3R and MC4R [55].

Since hBD1 is expressed in many tissues including brain, it was hypothesized that hBD1, which share the similar purported structure with hBD3, may interact with the neural MCRs involved in energy homeostasis [55]. Mouse and human β-defensin 1 are also shown to be micromolar MCR agonists [55]. We recently showed that cBD103ΔG23, the common variant of cBD103, increases ERK1/2 activation and cAMP accumulation when bound to the human or canine MC4R, suggesting that mutations in defensins may influence more than just the immune system [125].

### 3.2. Small Molecules

Numerous small molecule ligands for the MCRs have been developed due to their physiological importance [26,126,127,128,129,130]. The earliest work featured β-turn-inducing thioether-cyclized peptidomimetics [131]. In 2002, the first small molecule with single-digit nanomolar potency at the MC4R, THIQ, was described [132] (Figure 2). In the same year, a thioether cyclized peptidomimetic scaffold that generated one ligand, JB1n, possessing submicromolar potency at the MC4R, was reported [133]. Following these initial reports, several classes of small molecule ligands have been disclosed and reviewed in the literature [26,128,134,135].

MC4R was heavily targeted by the pharmaceutical industry due to its role in obesity [17]. Since MC4R agonists decrease food intake in rodent models [88], small molecule MC4R agonists were investigated as potential therapeutics to promote weight loss. However, administration of purported MC4R-selective compounds identified several side effects including hypertension, increased heart rate, skin darkening, and sexual arousal [136,137,138,139]. To avoid undesirable side effects, targeting of the centrally located MC3R, which also plays a role in food intake and fat disposition [18,19,140], has emerged as a valuably alternative approach. As recently reported, pyrrolidine bis-cyclic guanidines have nanomolar agonist potency at the MC3R and over 10-fold selectivity for the MC3R over the MC4R through “unbiased” mixture-based high-throughput screening approaches [141]. Recently, Ericson et al. (2017) [26] and Yeo et al. (2021) [135] published two outstanding reviews that summarized recent small molecule ligands of MCRs at humans and rodents.

Many new, potent, and enzyme-resistant analogs of melanocortin peptides have been developed based on the extensive studies of the melanocortin peptide, α-MSH. These agonists include NDP-α-MSH ([Nle4-D-Phe7]-α-MSH), melanotan II (MTII, Ac-Nle-c [Asp-His-DPhe-Arg-Trp-Lys]-NH_2_), and SHU9119 (Ac-Nle-c [Asp-His-Dnal(2′)-Arg-Trp-Lys]-OH). NDP-MSH has nanomolar to subnanomolar potencies in MC1R and MC3-5R (Table 2). Thirty-four years after its discovery, NDP-MSH was approved in the European Union as a treatment for adult erythropoietic protoporphyria in 2014 [142]. MTII is a nonselective melanocortin agonist with nanomolar to subnanomolar potencies [143,144] (Table 2). Since its discovery, MTII has been used as an in vitro and in vivo probe, with central ICV administration of MTII inhibiting food intake in mice [88]. SHU9119 is a nanomolar to subnanomolar agonist in MC1R and MC5R with antagonist activity in MC3R and MC4R [61] (Table 2). As the first peptide ligand discovered with potent antagonist activity at the MC3R and MC4R, ICV administration of SHU9119 was shown to significantly increase food intake in mice [88,140].

As in mammals, melanocortin system has also been shown to be involved in regulation of food intake and energy homeostasis in lower vertebrates such as fish (reviewed in [27]). In fact, the two neural MCRs, Mc3r [79,80,102,145] and Mc4r [104,105,106,107,109,110,111,146,147], have also been identified and characterized in several piscine species. Although MC3R is well studied in mammals, the studies on physiology and pharmacology of teleost Mc3rs are still limited, potentially due to the absence of *mc3r* genes in some teleosts, such as fugu, medaka, and stickleback [114,148,149].

MC4R also plays an important role in regulating energy homeostasis in teleosts. Previous in vivo studies with MC4R ligands showed MC4R agonist (MTII) suppresses food intake whereas the MC4R antagonist (SHU9119) increases the food intake in rainbow trout and goldfish after ICV administration [104,150]. In rainbow trout, ICV administration of MC4R specific antagonist (HS024) is more potent than MC3R/MC4R antagonist (SHU9119) in increasing food intake [150]. In addition, Mexican cavefish (*Astyanax mexicanus*) *mc4r* mutations contribute to physiological adaptations to nutrient-poor conditions by increasing appetite, growth, and starvation resistance [151]. The study of regulation of energy homeostasis in teleosts is related to better economic return in aquaculture, therefore the regulation of energy homeostasis by piscine neural Mc3r and Mc4r deserves further exploration. The large variations in genetics, ecology, morphology, anatomy, and physiology of teleost species result in complex species-specific energy homeostasis regulatory mechanisms [152]. For example, a large proportion of teleosts continue to grow during the entire life span, which is in contrast to the determinate growth in mammals and model animals, such as zebrafish [152]. Whether the melanocortin system functions differently in different species is unknown.

#### 3.2.1. THIQ

THIQ (Figure 2A), a tetrahydroisoquinolone ligand, is a synthetic small molecule compound developed by Merck. It was the first selective small molecule agonist of the MC4R with high affinity (IC_50_, 1.2 nM) and potency (EC_50_, 2.1 nM) [132]. Structure–function relationship study shows that THIQ competes with [Nle^4^,D-Phe^7^]-α-MSH (NDP-MSH) for binding to the MC4R, sharing several binding determinants with peptide agonists but also possessing unique binding determinants [153]. This small molecule MC4R agonist has been shown to attenuate brain inflammation and promote survival [154]. We showed that THIQ is a pharmacoperone of the MC4R, rescuing the cell surface expression and signaling of some intracellularly retained MC4R mutants [155].

THIQ binds to human MC4R orthosterically, but binds to fish Mc4r allosterically. In teleosts, including spotted scat (*Scatophagus argus*) [105], swamp eel (*Monopterus albus*) [107], and spotted sea bass (*Lateolabrax maculatus*) [109], THIQ fails to displace radiolabeled NDP-MSH but stimulates intracellular cAMP accumulation, suggesting that THIQ acts as an allosteric agonist in teleost Mc4r. THIQ exhibits dramatically different pharmacology at teleost and human MC4Rs, suggesting that for any potential use of small molecule ligands developed at human MC4R, pharmacological studies on Mc4r of the intended species need to be performed before any field trials in fish.

#### 3.2.2. Ipsen 5i

Ipsen 5i (Figure 2B), as the representative member of the amino-benzimidazole series, was first described by Roubert and colleagues at Ipsen with a Ki of 2 nM at the human MC4R [156]. Although Ipsen 5i has a high affinity with the MC4R, it has relatively low functional antagonist potency (77 nM) [156], minimizing its antagonizing effect on NDP-MSH. We also demonstrated that Ipsen 5i is a high-affinity inverse agonist of MC4R competing with NDP-MSH for binding to the MC4R while low affinity for the MC3R (Ki, 400 nM) [157]. Although Ipsen5i decreases basal signaling at the classical Gs-cAMP pathway, it acts as an agonist in the mitogen-activated protein kinase pathway [95]. We reported that Ipsen 5i is a novel potent pharmacoperone of the MC4R, correcting trafficking and signaling of a significant portion (73%) of intracellularly retained mutants [158,159].

A recent study with two teleost Mc4r showed that Ipsen 5i serves as a neutral allosteric modulator at cAMP signaling pathway but allosteric agonists at the ERK1/2 signaling pathway [103]. In this study, we showed that Ipsen 5i can bind to either spotted scat or grass carp Mc4r allosterically, failing to compete with radiolabeled NDP-MSH [103]. With swamp eel Mc4r, Ipsen 5i also binds to allosteric site(s) but induces an approximately two-fold increase in intracellular cAMP level, revealing it to be a weak agonist on swamp eel Mc4r [107]. Therefore, Ipsen 5i can have different effects on the Mc4r from different fish.

#### 3.2.3. ML00253764

The peptidomimetic compound ML00253764 (Figure 2C) was initially reported to function as an MC4R antagonist in reducing tumor-induced weight loss after peripheral administration with a low binding affinity (Ki, 160 nM) [160]. This was confirmed in another study using mice grafted with Lewis lung carcinoma tumors [161]. We subsequently showed that ML00253764 behaves as an inverse agonist at the Gs-cAMP pathway and meanwhile serves as an agonist in the mitogen-activated protein kinase pathway in wild-type and several constitutively active mutant human MC4Rs [93,95]. We recently showed that at spotted scat and grass carp Mc4r, ML00253764 serves as a neutral allosteric modulator at cAMP signaling but allosteric agonist at the ERK1/2 signaling pathway [103].

### 3.3. Pharmacoperones

Dysfunction in folding caused by genetic mutations in numerous genes causes protein conformational diseases [162]. Pharmacological chaperones (pharmacoperones, first coined by the late P. Michael Conn [163]), specifically bind to target protein, stabilizing the native conformation or facilitating the folding of nonnative intermediates into native conformation [164,165,166,167,168,169]. They can serve as novel therapeutics for treating genetic diseases caused by mutations in GPCR genes that result in misfolded mutant receptors [167,170,171,172,173,174,175,176,177]. So far, numerous ligands have been identified as pharmacoperones for GPCRs, such as SR121463A for arginine vasopressin V2 receptor [178], naltrexone for δ-opioid receptor [179], 11-*cis*-7-ring retinal for rhodopsin [180], Org 42599 for luteinizing hormone receptor [181], and IN3 for gonadotropin-releasing hormone receptor [182].

The MC4R is a critical regulator of energy homeostasis, regulating both food intake and energy expenditure [157]. Numerous *MC4R* mutations have been identified from 141,456 humans and have been registered in gnomAD v2.1.1 database (https://gnomad.broadinstitute.org/, accessed on 25 August 2022) [183,184]. Many functional studies of naturally occurring *MC4R* mutations showed that the most common defect is decreased or lack of cell surface expression [157,185].

ML00253764, the first pharmacoperone for MC4R, was identified by our team in 2009 [157,186]. Since then, seven more antagonists have been identified as pharmacoperones for the MC4R: MTHP, PPPone, MPCI, DCPMP, NBP, Ipsen 5i, Ipsen 17, and UM0130866 [158,159,176,187,188]. Antagonist-rescued proteins arrive at the plasma membrane in an antagonist-bound state, potentially preventing the receptor being activated by endogenous agonists. High-affinity antagonist pharmacoperones are generally difficult to dissociate from their target receptors, while low-affinity antagonist pharmacoperones are easy to dissociate. In contrast to antagonist pharmacoperones, agonist pharmacoperones do not need to dissociate from the target receptor. The small molecule potent agonist of the MC4R, THIQ, is also identified as a pharmacoperone, promoting the proper folding and subsequent signaling of intracellularly retained mutant MC4Rs [155]. Interestingly, all the nine antagonist and one agonist pharmacoperones of the MC4R increase the cell surface expression of the wild-type MC4R, indicating pharmacoperone therapy could be used to treat obese patients without any *MC4R* mutation [157,158,186,187]. Very recently, Bouvier’s group showed that a pharmacoperone can rescue the mutant MC4R, R165W, in knockin mice [176].

### 3.4. Ligands Identified from Animals

It was suggested that studying nature-derived peptides confer advantages for developing GPCR ligands as drug discover effort [189]. Full agonists were identified from frog peptides that can bind to human MC4R with good affinities, although EC_50_s are in the tens of micromolar range [190].

Animal venoms contain numerous chemicals with diverse biological activities. In a bank of 3597 toxins, 8% are found to interact with human MC4R [191]. Two peptides, lacking HFRW pharmacophore, were described that can bind to the four MCRs (except MC2R) with low micromolar affinities and activate the human MC1R, causing increased intracellular cAMP [191]. They can work synergistically with α-MSH, although the mechanism is not clear. These peptides are not related structurally to the endogenous agonists, the melanocortins, but have some similarities with AgRP and hBD3 [191]. Further characterization and optimization of these toxins might identify new tools for the study of diverse aspects of MCR pharmacology, such as biased signaling and high selectivity, potentially leading to new therapeutics.

## 4. Ligands Used in Human Medicine

### 4.1. ACTH

ACTH is an old forgotten melanocortin peptide that might potentially be rescued and repurposed for new indications [192]. ACTH received the US Food and Drug Administration (FDA) approval for use in humans in 1952, only three years after it was first tested in rheumatoid arthritis patients [193]. Philip S. Hench, Edward C. Kendall, and Tadeus Reichstein were awarded the Nobel Prize in 1950 in Physiology or Medicine for these discoveries on ACTH and adrenal hormones. However, highly efficient methods for glucocorticoid synthesis were subsequently developed and oral forms also became available, making glucocorticoid the treatment of choice to the detriment of ACTH. The use of this melanocortin peptide becomes very limited, usually regarded as a second choice when glucocorticoid therapy is not possible.

ACTH was the gold standard therapy for multiple sclerosis in the 1960s, but its use decreased with the advent of glucocorticoid [194]. H.P. Acthar^®^ Gel is the only treatment available for infantile spasm, a medical condition usually diagnosed within the first year of life, consisting of seizures and mental retardation with poor prognosis [195,196]. A retrospective study on 181 gout patients reported positive response in 77.9% of patients within one day after ACTH injection, indicating that it is effective and safe for the treatment of gout [197,198]. The efficacy of ACTH in the treatment of nephrotic syndrome is well documented, as it improves proteinuria, reduces kidney inflammation, and corrects dyslipidemia, which could not be fully explained by induction of glucocorticoids [199]. It is worth noting that the extra-adrenal effects of ACTH were not known when it was first commercialized, and the relevance of these effects is only emerging recently [199,200,201].

Interestingly, exactly 50 years after ACTH was approved by the FDA, Getting et al. envisaged a novel mechanism of action and discovered that the anti-inflammatory effects of ACTH persist in adrenalectomized rats using a model of knee gout [202]. ACTH also reduces neutrophil infiltration in a model of crystal inflammation and decreases the production of the chemoattractant cytokine CXCL-1, an effect blocked when the MC3R/MC4R antagonist SHU9119 was used [202,203]. This cortisol-independent effect is mediated by the MC3R, which is expressed in immune cells and the brain, presenting this receptor as a novel therapeutic target for ACTH-like drugs devoid of cortisol-related side effects. These findings have renewed interest in ACTH therapy by reconsidering the use of ACTH for new indications and proposing innovative therapeutic targets like the melanocortin system for the development of new anti-inflammatory therapies [192].

In addition, the ability of ACTH to directly modulate local CNS inflammation has been reported in several in vitro studies. Using rat brain cultures containing oligodendroglia, astrocytes, and microglia, preincubated with cytotoxic agents, ACTH was shown to protect mature oligodendroglia and oligodendroglia progenitor cell from death induced by staurosporine, kainate, quinolinic acid, or reactive oxygen species [204,205]. Using rat glial cultures, the same group demonstrated that ACTH induces proliferation of oligodendroglia progenitor cells and accelerates differentiation of platelet-derived growth factor receptor-α to a later stage characterized by greater expansion of oligodendroglia myelin-like sheets compared to untreated cells [205]. Furthermore, the same group showed that ACTH also protects cultured rat forebrain neurons from excitotoxic, apoptotic, oxidative and inflammation related insults [206], but the specific MCR subtype(s) involved are not known. Since excitotoxic damage to neurons is an important cause to several experimental and clinical CNS diseases, it is reasonable to explain the therapeutic benefits of ACTH in several inflammatory animal models of CNS disorders [206]. However, the contribution from direct effects on oligodendroglia, oligodendroglia progenitor cell, or neurons within the brain and whether such beneficial actions can be observed in vivo in human patients remain to be investigated.

### 4.2. Setmelanotide

Obesity has reached epidemic proportions in developed countries and represents a major public health challenge. The pandemic of obesity is growing at an alarming rate, with an estimated direct and indirect cost of $150 billion per year in the US alone, and one out of five deaths is linked in one way or another to obesity [207]. Obesity is a main risk factor for many diseases, such as type II diabetes, cardiovascular diseases, fatty liver, and cancer, among others [208,209]. For many obese individuals, behavioral and lifestyle modifications are insufficient for long-term weight-loss maintenance and thus depend on pharmacological interventions that alter either appetite or absorption of calories. Until 2014, FDA-approved weight loss medications for obesity management include orlistat, lorcaserin, liraglutide, bupropion-naltrexone, and phentermine-topiramate for long-term treatment; and benzphetamine, phentermine, phendimetrazine, and diethylpropion for short-term use [210].

Over the past few decades, concerted efforts have been made to translate melanocortin ligands into clinical therapies [26,135]. Both α- and β-MSHs were injected into humans in 1961 [44]. Since then, several oral small molecules and injectable peptide agonists of MC4R have been clinically evaluated for treating obesity, including oral small molecule MK-0493 (Merck) [211] and peptide agonists LY2112688 (Eli Lilly) [138], MC4-NN-0453 (Novo Nordisk) [212], AZD2820 (Astra Zeneca) [213], and setmelanotide (Rhythm Pharmaceuticals) [214,215,216,217,218]. However, most MC4R agonist drug candidates developed as potential therapeutics for treating obesity failed in the clinic despite preclinical efficacy, due to lack of efficacy and side effects. The exception up to now has been setmelanotide (Figure 3A), a cyclic octapeptide that has a high potency for the two neutral MCRs (EC_50_ at MC4R is 0.032 nM and EC_50_ at MC3R is 0.69 nM) [57]. This synthetic peptide was approved by FDA at the end of 2020 for treating obesity caused by genetic defects in pro-opiomelanocortin (*POMC*), leptin receptor (*LEPR*), or proprotein convertase subtilisin/kexin type 1 (*PCSK1*) [219].

In a 28-day Phase 1b clinical trial, setmelanotide led to moderate weight loss in obese people with *MC4R* deficiency [220]. The efficacy of setmelanotide was more pronounced in obese subjects with *POMC* and LEPR deficiency, with 80% of *POMC*-deficient subjects and 45% of *LEPR*-deficient subjects losing >10% of their weight within approximately one year [215,217]. In addition, compared to other clinically tested MC4R agonists, setmelanotide does not elicit undesirable side effects such as cardiovascular effects, with only minor adverse events including hyperpigmentation, nausea/vomiting, penile erection, and injection site reactions [214,215,217].

The safety of setmelanotide might be caused by differential signaling pathways [215,221]. Setmelanotide, as a mimetic of α-MSH, initiates similar signaling as α-MSH in Gαs and β-arrestin 2 pathways [222]. Setmelanotide also potently activates Gαq signaling through MC4R, which cannot be efficiently antagonized by AgRP [215,223]. It was found that 100 nM of AgRP cannot antagonize setmelanotide-stimulated phospholipase C activation (using the NFAT reporter gene assay) while stimulation by α-MSH is antagonized [215]. The MC4R-mediated Gαq signaling has been shown to reduce food intake without affecting cardiovascular activity [224]. Therefore, the biased ligands for MC4R that selectively activate Gαq signaling to decrease food intake but not trigger Gαs signaling with cardiovascular side effects might be better drug candidates for obesity treatment [21].

### 4.3. Bremelanotide (PT-141)

In addition to obesity therapies, the other potential use of MC4R agonists as therapeutics for erectile dysfunction and premenopausal hypoactivity sexual desire disorder has also been investigated, including melanotan II (University of Arizona) [225], bremelanotide (Palatin Inc) [226], and PF-00446687 (Pfizer Inc) [227].

Bremelanotide (PT-141) (Figure 3B) is a MC4R agonist approved in June of 2019 by the FDA for erectile dysfunction in men and hypoactivity sexual desire disorder in premenopausal women, as characterized by low sexual desire that causes marked distress or interpersonal difficulty [228,229]. Bremelanotide was developed by Palatin Technologies (trade name Vyleesi) as a self-administered and on-demand subcutaneous therapy for female sexual dysfunction [230]. MC4R, expressed on neurons in many areas of the central nervous system and some peripheral tissues, are thought to be the most important for sexual function [139,231,232]. As a synthetic peptide analogue of α-MSH, bremelanotide has a high potency for the MC4R (EC_50_ of 0.25 nM), making it possible to modulate brain pathways involved in sexual response [232,233].

The most common adverse events of bremelanotide are nausea, flushing and headache, with no marked dosage dependence [226,234]. Meanwhile, bremelanotide has limited drug–drug interactions, including no clinically significant interactions with alcohol [235]. The only other FDA-approved treatment indicated for hypoactivity sexual desire disorder is flibanserin, which was approved in 2015. Recently, clinical practices revealed that bremelanotide may offer advantages over flibanserin for certain individuals [235]. Nevertheless, further studies are needed to review the use of this drug in menopausal and postmenopausal populations and to determine its place in therapy of these patients.

### 4.4. Drug Repurposing

MC3R and MC4R are inextricably linked to obesity. However, research on the old drugs repurposing for new therapeutic uses for the melanocortin system is scarce. The FDA-approved drug fingolimod, a sphingosine 1-phosphate receptor agonist, is used in monotherapy for the treatment of relapsing-remitting multiple sclerosis [236,237]. Recently, fingolimod is demonstrated to exert antiangiogenic activity in diabetic retinopathy not only through the sphingosine 1 phosphate receptor but also through activating MC1R and MC5R [238]. However, in the study, no binding and signaling data were provided; only in silico analysis was done; and AgRP was used as MC1R antagonist [238], which is indeed an antagonist for the MC3R and MC4R. Pharmacology data from cells expressing MC1R and MC5R are needed to confirm the conclusions reached in this study.

Another case is the marketed drug fenoprofen. Long known for the antiarthritic activities and efficacy in rheumatoid arthritis and osteoarthritis, fenoprofen is used as a nonsteroidal anti-inflammatory drug and inhibits cyclooxygenase [239]. Montero-Melendez et al. demonstrated that fenoprofen displays antiarthritic actions on cartilage integrity and synovitis mediated by MC3R, with fenoprofen serving as a positive allosteric modulator, revealing an important contribution of MC3R in the therapeutic actions of fenoprofen in arthritis [240]. In agreement with this study, we showed that positive allosteric modulator activity and biased signaling induced by fenoprofen are observed not only at wild-type MC3R, but also at naturally occurring MC3R mutants, suggesting that this biased allosteric enhancer action might constitute a novel therapeutic opportunity for obese patients harboring these mutations [241].

Another recent study suggested that metformin, used as first-line treatment for type 2 diabetes mellitus, inhibits the activation of MC2R and MC3R [242]. At the MC3R, metformin does not change the sensitivity to α-MSH stimulation, but decreases the maximal response. Since no binding experiment was done, it is not clear whether metformin binds to the MC3R orthosterically or allosterically.

## 5. Conclusions

Numerous melanocortin ligands have been developed in the 70 years since the sequences of the first endogenous ligands were elucidated. While much of the early focus was on the development of compounds that alter pigmentation, the cloning and structure analysis of the MCRs led to the development of potent and selective ligands. Targeting neural MCRs (MC3R and MC4R) is emerging as a therapeutic approach for metabolic diseases and chronic inflammation. This review summarized the advances in the ligands for neural MCRs, including classical endogenous ligands, nonclassical ligands such as small molecules and pharmacoperones, and some ligands approved for human medicine. Development of novel nonclassical ligands and old drugs repurposed for targeting the neural MCRs to treat obesity and related diseases are of imminent importance to fulfill a highly unmet medical need in the future.

## Figures and Tables

**Figure 1 biomolecules-12-01407-f001:**
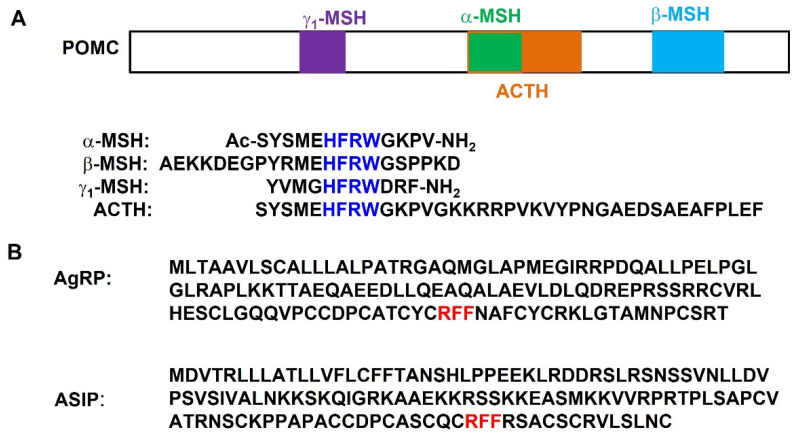
Endogenous ligands for melanocortin receptors. (**A**) Schematic diagram of POMC and sequences of four agonists; (**B**) sequences of two antagonists. The HFRW motif is shown in blue. The RFF motif is highlighted in red.

**Figure 2 biomolecules-12-01407-f002:**
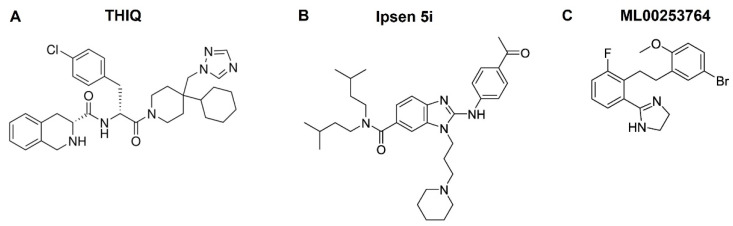
Chemical structures of THIQ (**A**), Ipsen 5i (**B**), and ML00253764 (**C**).

**Figure 3 biomolecules-12-01407-f003:**
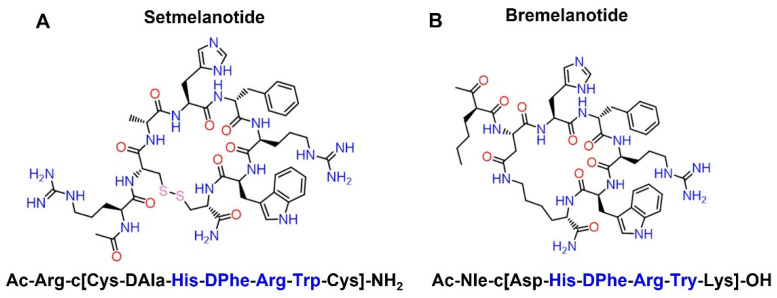
Amino acid sequences and chemical structures of setmelanotide (**A**) and bremelanotide (**B**). The functional motif is shown in blue.

**Table 1 biomolecules-12-01407-t001:** Pharmacological properties of endogenous ligands at human melanocortin receptors.

Ligands	Binding Affinity [28,29]	Activity [30]
MC1R	α-MSH > β-MSH > γ_3_-MSH > ACTH > γ_1_-MSH > γ_2_-MSH	γ_1_-MSH > α-MSH > γ_2_-MSH > γ_3_-MSH > β-MSH
MC3R	γ_1_-MSH > γ_3_-MSH > β-MSH > γ_2_-MSH > α-MSH > ACTH	γ_2_-MSH > γ_3_-MSH > γ_1_-MSH > α-MSH > β-MSH
MC4R	β-MSH > α-MSH > ACTH > γ_1_-MSH > γ_3_-MSH > γ_2_-MSH	α-MSH > β-MSH > γ_2_-MSH > γ_1_-MSH > γ_3_-MSH
MC5R	α-MSH > β-MSH > ACTH > γ_1_-MSH > γ_2_-MSH= γ_3_-MSH	α-MSH > β-MSH > γ_1_-MSH > γ_2_-MSH > γ_3_-MSH

**Table 2 biomolecules-12-01407-t002:** Binding affinities of melanocortin ligands at human melanocortin receptors.

Peptides	MC1R	MC3R	MC4R	MC5R	Index	Reference
α-MSH	0.0334	20.7	641	8240	K_i_ (nM)	[28]
	0.23	31.5	900	7160	K_i_ (nM)	[46]
	0.32	15.5	41.4	332	K_i_ (nM)	[30]
	0.94	27.74	30.37	125.3	K_i_ (nM)	[47]
	1.5	46	26	150	K_i_ (nM)	[48]
	3.9	19	19	120	IC_50_ (nM)	[49]
	5.97	50.04	38.7	557	IC_50_ (nM)	[50]
β-MSH	1.17	13.4	376	14,400	K_i_ (nM)	[28]
	0.89	10.64	8.18	76.9	K_i_ (nM)	[47]
	0.864	23.2	19.9	306	K_i_ (nM)	[30]
γ_1_-MSH	2.68	7.06	29,000	42,600	K_i_ (nM)	[28]
	2.75	11	1300	552	K_i_ (nM)	[30]
γ_2_-MSH	11.2	17.7	>100,000	>100,000	K_i_ (nM)	[28]
	20.8	57.2	6250	3250	K_i_ (nM)	[30]
γ_3_-MSH	1.39	10.9	33,500	>100,000	K_i_ (nM)	[28]
	0.419	5.84	117	336	K_i_ (nM)	[30]
ACTH	2.5	86.9	693	17,000	K_i_ (nM)	[28]
AgRP	-	11.2	9	25.6	IC_50_ (nM)	[51]
ASIP	23	195	70	>1000	K_i_ (nM)	[52]
	0.47	6.4	0.14	1.16	K_i_ (nM)	[53]
Lipocalin 2	86.96	82.13	51.39	-	K_d_ (nM)	[54]
hBD1	7000		>100,000		IC_50_ (nM)	[55]
hBD3	26,000		>100,000		IC_50_ (nM)	[55]
	42		110		K_i_ (nM)	[56]
Setmelanotide	3.9	10	2.1	430	K_i_ (nM)	[30]
	5.5	18	5	1910	IC_50_ (nM)	[57]
Bremelanotide	3.4	220	29	190	IC_50_ (nM)	[57]
NDP-MSH	0.046	0.78	0.30	0.48	K_i_ (nM)	[48]
	0.109	0.680	0.620	0.891	K_i_ (nM)	[30]
MTII	0.2	20	1.5	23	IC_50_ (nM)	[57]
	0.27	24	2.66	23.1	K_i_ (nM)	[30]
	0.36	3.1	0.18	1.2	IC_50_ (nM)	[58]
	0.686	34.1	6.6	46.1	K_i_ (nM)	[46]
	6.4	176	0.25	17	K_i_ (nM)	[48]
SHU9119	0.62	0.23	0.07	0.065	IC_50_ (nM)	[58]
	0.714	1.2	0.36	1.12	K_i_ (nM)	[46]

**Table 3 biomolecules-12-01407-t003:** Functional pharmacology of melanocortin ligands at human melanocortin receptors.

Peptides	MC1R		MC3R		MC4R		MC5R		Reference
	EC_50_ (nM)	pA_2_	EC_50_ (nM)	pA_2_	EC_50_ (nM)	pA_2_	EC_50_ (nM)	pA_2_	
α-MSH	0.057		0.669		0.21		0.807		[31]
	1.01		1.04		4.7		10.5		[30]
	3.4		1.1		1.9		16		[58]
β-MSH	4.24		1.59		6.62		23.7		[30]
γ_1_-MSH	0.932		0.734		48.5		101		[30]
γ_2_-MSH	3.05		0.487		93.7		500		[30]
	-		1		55		200		[59]
γ_3_-MSH	1.34		0.465		42.2		233		[30]
AgRP				8.4		8.6			[51]
ASIP		9.3		8.2		9.9		8.9	[53]
Lipocalin 2	1.52		1.83		1.41				[54]
hBD1	7400		>100,000		21,000		>100,000		[55]
hBD3	400		35% @100 μM		2600		45% @100 μM		[55]
Setmelanotide	5.8		5.3		0.27		1600		[30]
	0.26		0.69		<0.032		-		[57]
Bremelanotide	0.095		2.4		0.25		-		[57]
NDP-MSH	0.462		0.109		0.075		0.253		[30]
MTII	0.2		0.51		0.05		5.33		[30]
	0.3		1.3		2.9		3.3		[60]
	0.32		1.1		0.26		2.3		[58]
SHU9119	0.036		Partial agonist	8.3		9.3	0.434		[61]

## Data Availability

The raw data supporting the conclusions of this article will be made available by the authors upon request, without undue reservation.

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
