# Peer review of "Ligands for Melanocortin Receptors: Beyond Melanocyte-Stimulating Hormones and Adrenocorticotropin"

_biomolecules, 2022, doi:10.3390/biom12101407_

Round 1

Reviewer 1 Report

The authors provide the most extensive review that I have seen on the peptide and non-peptide ligands that can have pharmacological effects on the melanocortin-3 receptor and the melanocortin-4 receptor.

The organization of the review is logical. The text flows very well and the authors in general wrote each section in a concise manner. The appropriate references have been cited, and the figures are journal quality.

I have only two recommendations.

On line 45 I would suggest the phrase, "...based on the chronological order of their characterization..."

Line 50 I would suggest "feeding" rather than "feed".

Author Response

The authors provide the most extensive review that I have seen on the peptide and non-peptide ligands that can have pharmacological effects on the melanocortin-3 receptor and the melanocortin-4 receptor.

The organization of the review is logical. The text flows very well and the authors in general wrote each section in a concise manner. The appropriate references have been cited, and the figures are journal quality.

Thank you for your positive comments.

I have only two recommendations.

On line 45, I would suggest the phrase, "...based on the chronological order of their characterization..."

Response: This sentence has been revised as “In the early 1990s, with degenerate PCR cloning techniques, three additional related MCRs were cloned, named MC3R, MC4R, and MC5R, based on the chronological order of their characterization, with MSH receptor renamed MC1R and ACTH receptor renamed MC2R”.

Line 50 I would suggest "feeding" rather than "feed"

Response: The “feed” was replaced by “feeding”.

Reviewer 2 Report

In this article, the authors reviewed the discovery and identification of lots of ligands for neural MCRs and from endogenous peptides 70 years ago and the development of therapeutic approach for treating metabolic monogenic obesity. Most of the ligands for MC3R and MC4R are well introduced and elaborated through which the readers could be inspired.

1. Not the structural nor functional features but the history of these molecular are well introduced. For a reader who is interested in the peptides or molecules binding to neural MCRs (as the authors stated in the abstract), some features, such as the binding configuration, pharmacological modification, or comparative studies between different ligands toward a specific receptor, will be more helpful. The authors should focus more on this aspect to revise the manuscript.

2. It is hard to ignore the logical gaps throughout the entire article, as well as the related pharmacological studies on fish. An evolutionary perspective is inspired but should be arranged in the right place. As an example, in lines 217-232, in this section 3.2 small molecules, it is weird to jump into piscine species only because MTII and SHU9119 are functional in rainbow trout and goldfish. However, important roles of MTII and SHU9119 in mammals should also be mentioned.

3. Tables listing each category of ligands and their molecular features such as EC50/IC50, functional motif, binding sites, and related references will be very helpful for readers to understand these complicated ligand. Also comparative analysis could also be presented in these tables.

4. Lines 151-157, this paragraph mainly discussed GHSR and LEAP-2, was somehow irrelevant to this topic.

5. Lines 282-311, also, the author focused too much on introducing the concept of the pharmacological chaperone.

6. Section 4.1, whether ACTH stimulates MC3R in CNS is not well established. This section is suitable for a review of ACTH, but not ligands for MCRs.

7. Lines 473-501 are not relevant to the main topic.

8. The entire manuscript should be rearranged to get a better logical connections between different sections.

Author Response

In this article, the authors reviewed the discovery and identification of lots of ligands for neural MCRs and from endogenous peptides 70 years ago and the development of therapeutic approach for treating metabolic monogenic obesity. Most of the ligands for MC3R and MC4R are well introduced and elaborated through which the readers could be inspired.

  1. Not the structural nor functional features but the history of these molecular are well introduced. For a reader who is interested in the peptides or molecules binding to neural MCRs (as the authors stated in the abstract), some features, such as the binding configuration, pharmacological modification, or comparative studies between different ligands toward a specific receptor, will be more helpful. The authors should focus more on this aspect to revise the manuscript.

Response: We have added Tables 1 to 3 on these aspects.

  1. It is hard to ignore the logical gaps throughout the entire article, as well as the related pharmacological studies on fish. An evolutionary perspective is inspired but should be arranged in the right place. As an example, in lines 217-232, in this section 3.2 small molecules, it is weird to jump into piscine species only because MTII and SHU9119 are functional in rainbow trout and goldfish. However, important roles of MTII and SHU9119 in mammals should also be mentioned.

Response: The logical gaps have been revised throughout the entire manuscript, as well as the related pharmacological studies on fish. For instance, the content about fish AgRP has been modified as “As in mammals, AgRP are also determined to play critical roles in regulating food intake and energy homeostasis in teleosts. And this protein still acts as an inverse agonist at neural melanocortin receptors of many fish species. We and others have shown that both Mc3r and Mc4r in fishes have very high constitutive activities, compared with human MC3R and MC4R, respectively. But unlike mammals, AgRP has been identified in several fish species (eg, goldfish Carassius auratus, zebrafish Danio rerio, Atlantic salmon Salmo salar, seabass Dicentrarchus labrax, and pufferfish Takifugu rubripes) with two gene products (AgRP1 and AgRP2). In seabass (Perciforme), long-term fasting increases hypothalamic expression of AgRP1 but decreases that of AgRP2, suggesting an isoform-specific orexigenic action in fish”.

Lines 217-232, in this section 3.2 small molecules, has been added transitional sentences to get better logical connections, such as “Also, MC4R plays an important role in regulating energy homeostasis in teleosts” and “The large variations in genetics, ecology, morphology, anatomy and physiology of teleost species result in complex species-specific energy homeostasis regulatory mechanisms. For example, a large proportion of teleosts continue to grow during the whole life span, which is in contrast with the determinate growth in mammals and model animals, such as zebrafish”.

At last, important roles of MTII and SHU9119 in mammals have been added to the manuscript, as “Many new, potent, and enzyme-resistant analogs of melanocortin peptides have been developed based on the extensive studies of the melanocortin peptide, α-MSH. These agonists include NDP-a-MSH ([Nle4-D-Phe7]-α-MSH), Melanotan II (MTII, Ac-Nle-c [Asp-His-DPhe-Arg-Trp-Lys] -NH2), and SHU9119 (Ac-Nle-c [Asp-His-Dnal(2’)-Arg-Trp-Lys] -OH). NDP-MSH has nanomolar to sub-nanomolar potencies in MC1R and MC3-5R (Table 2). Thirty-four years after its discovery, NDP-MSH was approved in the European Union as a treatment for adult erythropoietic protoporphria in 2014. MTII is a non-selective melanocortin agonist with nanomolar to subnanomolar potencies (Table 2). Since its discovery, MTII has been used as an in vitro and in vivo probe, with central icv administration of MTII inhibiting food intake in mice. SHU9119 is a nanomolar to sub-nanomolar agonist in MC1R and MC5R with antagonist activity in MC3R and MC4R (Table 2). As the first peptide ligand discovered with potent antagonist activity at the MC3R and MC4R, icv administration of SHU9119 was shown to significantly increase food intake in mice”.

  1. Tables listing each category of ligands and their molecular features such as EC50/IC50, functional motif, binding sites, and related references will be very helpful for readers to understand these complicated ligand. Also comparative analysis could also be presented in these tables.

Response: See Response to Comment 1 above. The binding affinities (presented as Ki and IC50) and functional pharmacology (EC50 and pA2) of melanocortin ligands have been added into Table 2 and 3 following the suggestion of the reviewer. The molecular features of each ligand such as functional motif and binding sites have been highlighted in the figures (Fig. 1 and Fig. 3) or illustrated in the revised manuscript, such as “The endogenous agonists of neural MCRs are a-, b-, and g-MSH, as well as ACTH, with the conserved tetrapeptide sequence His-Phe-Arg-Trp (HFRW) as the pharmacophore (Fig. 1A), the minimally active sequence that possesses agonist activity in the classic frog and lizard skin bioassays. Activation of the MCRs with MSH analogous indicated that the length of ACTH or MSH is different for a specific receptor subtype. Four amino acids (His-DPhe-Arg-Trp) are the minimal MSH peptide for MC1R activation. First 17 amino acid of ACTH is required for MC2R activation. Four amino acids (His-DPhe-Arg-Trp) in MSH are required for MC3R and MC5R activation. Three amino acids (DPhe-Arg-Trp) in MSH are required for MC4R activation. Of the four major endogenous melanocortins, the human MC1R and MC5R have the highest affinities for a-MSH, the MC3R has the highest affinity for g1-MSH; the MC4R has the highest affinity for b-MSH (Table 1)”.

  1. Lines 151-157, this paragraph mainly discussed GHSR and LEAP-2, was somehow irrelevant to this topic.

Response: We have deleted this paragraph as suggested.

  1. Lines 282-311, also, the author focused too much on introducing the concept of the pharmacological chaperone.

Response: We have simplified the concept of the pharmacological chaperone according to the reviewer's suggestion, as “Dysfunction in folding caused by genetic mutations in numerous genes causes protein conformational diseases. Pharmacological chaperones (pharmacoperones, first coined by P. Michael Conn), specifically bind to target protein, stabilizing the native conformation or facilitating the folding of non-native intermediates into native conformation. They can serve as novel therapeutics for treating genetic diseases caused by mutations in GPCR genes that result in misfolded mutant receptors. So far, numerous ligands have also been identified as pharmacoperones for GPCRs, such as SR121463A for arginine vasopressin V2 receptors (AVPR2), naltrexone for δ-opioid receptor, 11-cis-7-ring retinal for rhodopsin, Org 42599 for luteinizing hormone receptor, and IN3 for gonadotropin-releasing hormone receptor”.

  1. Section 4.1, whether ACTH stimulates MC3R in CNS is not well established. This section is suitable for a review of ACTH, but not ligands for MCRs.

Response: We have modified the section as suggested by the reviewer. Here, we streamlined the content of ACTH in human medicine. According to the question of whether ACTH stimulates MC3R in CNS, a new paragraph has been added to the manuscript, as “In addition, the ability of ACTH to directly modulate local CNS inflammation has been reported in several in vitro studies. Using rat brain cultures containing oligodendroglia, astrocytes, and microglia preincubated with cytotoxic agents, ACTH was shown to protect mature oligodendroglia and oligodendroglia progenitor cell from death induced by staurosporine, kainate, quinolinic acid, or reactive oxygen species. Using rat glial cultures, the same group demonstrated that ACTH induces proliferation of oligodendroglia progenitor cell and accelerates differentiation of platelet-derived growth factor receptor-a to a later stage characterized by greater expansion of oligodendroglia myelin-like sheets compared to untreated cells. Furthermore, the same group showed that ACTH also protects cultured rat forebrain neurons from excitotoxic, apoptotic, oxidative and inflammation related insults, but the specific MCR subtypes involved are not known. Since excitotoxic damage to neurons is an important cause to several experimental and clinical CNS diseases, it is reasonable to explain the therapeutic benefits of ACTH in several inflammatory animal models of CNS disorders. However, the contribution from direct effects on oligodendroglia, oligodendroglia progenitor cell, or neurons within the brain and whether such beneficial actions can be observed in vivo in human patients remain to be investigated”.

  1. Lines 473-501 are not relevant to the main topic.

Response: We have deleted the two paragraphs in order to make the structure of the manuscript more reasonable and fluent.

  1. The entire manuscript should be rearranged to get a better logical connection between different sections.

Response: We have modified the entire manuscript following the suggestion of the reviewer.

Round 2

Reviewer 2 Report

The revised manuscript has addressed all my comments and met the standard for publication.

Minor comments: 

Please go through the whole text to remedy the spelling and grammar issues.

for example:

line 38, Family should be "family".

line 49,serve should be "served".

line 68, MSHs should be up-right.